

# Paleo±Dust: Quantifying uncertainty in paleo-dust deposition across archive types

Nicolás J. Cosentino[1,2], Gabriela Torre[3,4], Fabrice Lambert[1], Samuel Albani[2], François De Vleeschouwer[5], Aloys Bory[6]

[1]Instituto de Geografía, Facultad de Historia, Geografía y Ciencia Política, Pontificia Universidad Católica de Chile, Macul, 7820436, Chile
[2]Dipartimento di Scienze dell'Ambiente e della Terra, Università degli Studi di Milano-Bicocca, Milan, 20126, Italy
[3]Facultad de Ciencias Exactas, Físicas y Naturales, Universidad Nacional de Córdoba, Córdoba, X5000JJC, Argentina
[4]Centro de Investigaciones en Ciencias de la Tierra (CICTERRA), Consejo Nacional de Investigaciones Científicas y Tecnológicas (CONICET), Cordoba, X5016GCA, Argentina
[5]Instituto Franco-Argentino para el Estudio del Clima y sus Impactos (IFAECI, UMI, CNRS-CONICET-UBA-IRD), Buenos Aires, C1428, Argentina
[6]Laboratoire d'Océanologie et de Géosciences (LOG, UMR 8187, Université de Lille-CNRS-Université Côte d'Opale-IRD), Lille, F-59000, France

*Correspondence to*: Nicolás J. Cosentino (nicolas.cosentino@uc.cl)

**Abstract.** Mineral dust aerosol concentrations in the atmosphere varied greatly on glacial-interglacial timescales. The greatest changes in global dust activity occurred in response to changes in orbital parameters that affect dust emission intensity through glacial activity, and dust lifetime in the atmosphere through changes in the global hydrological cycle. Long-term changes in surface dust deposition rate are registered in geological archives such as loess, peats, lakes, marine sediments, and ice. Data provided by these archives is crucial for guiding simulations of dust, and for better understanding the natural global dust cycle. However, the methods employed to derive paleo-dust deposition rates differ markedly between archives and are subject to different sources of uncertainty. Here, we present Paleo±Dust, an updated compilation of bulk and <10-µm paleo-dust deposition rate with quantitative 1-σ uncertainties that are inter-comparable among archive types. Paleo±Dust incorporates a total of 284 pre-industrial Holocene (pi-HOL) and 208 Last Glacial Maximum (LGM) dust flux constraints from studies published until December 2022, including for the first time peat records. We also recalculate previously published dust fluxes to exclude data from the last deglaciation and thus obtain more representative constraints for the last pre-industrial interglacial and glacial end-member climate states. Based on Paleo±Dust, the global LGM:pi-HOL ratio of <10-µm dust deposition rate is 3.1 ± 0.8 (1σ). We expect Paleo±Dust to be of use for future paleoclimate dust studies and simulations using Earth system models of high to intermediate complexity.

## 1 Introduction

Mineral dust aerosols interact with climate through multiple mechanisms at different timescales, constituting a long recognized, relevant player in the Earth system. However, although some of the dust-climate interactions are well understood





in principle, many remain relatively unconstrained quantitatively, to the point where it is still unknown whether the net

radiative effect of dust implies net cooling or warming in the present day (Kok et al., 2023). Thus, dust remains an important

contributor to past and future climate change uncertainty (e.g., Andreae et al., 2005; Sherwood et al., 2020).

Both natural and anthropogenic processes are responsible for present-day dust emissions (Ginoux et al., 2012; Stanelle et al., 2014; Chen et al., 2018). To constrain naturally induced changes in the dust cycle (a prerequisite to disentangle the present-day, human-induced component) one can turn to pre-industrial records of dust activity. Based on such records spanning the

last several hundreds of thousands of years, it has been shown that the most dramatic changes in dust activity have taken place at glacial-interglacial timescales (Lambert et al., 2008; Simonsen et al., 2019; Struve et al., 2022), although significant variability is present at all timescales (Lovejoy and Lambert, 2019). Thus, global observational datasets of past dust activity for the pre-industrial Holocene and last glacial periods are key to constraining natural processes involved in dust-climate interactions, through multiproxy comparisons, their use in guiding Earth system simulations of past climates, and as a tool

for model-data comparisons.

Paleo-dust archives include continental (i.e., loess-paleosol sequences, lake, peat, and ice) and marine sediments. The dust dynamic parameter that may be calculated from information preserved in these deposits is dust deposition rate, from which other aspects of the dust cycle must be inferred, such as atmospheric dust concentrations and dust emission rate, typically through Earth system simulations. Each archive type has its strengths and weaknesses in terms of spatial representativity and

how well they preserve dust (Albani et al., 2015), and it is only through a combination of constraints from all these archives that a spatially extensive coverage of paleo-dust activity can be achieved. The degree of coverage is important due to short-scale variability in dust emissions associated with the uneven distribution of dust sources, as well as to spatially abrupt changes in dust deposition fluxes associated with precipitation-controlled wet deposition and distance-to-dust-sources-controlled dry deposition.

Several compilations of paleo-dust deposition fluxes exist that combine constraints from different archive types (Mahowald et al., 1999, 2006; Kohfeld and Harrison, 2001; Tegen et al., 2002; Maher et al., 2010; Albani et al. 2014, 2015; Lambert et al., 2015), as well as ones that specialize in marine sediment cores (Kohfeld et al., 2013; Kienast et al., 2016). These compilations have allowed data-model comparisons (Hopcroft et al., 2015; Kienast et al., 2016; Ohgaito et al., 2018; Otto-Bliesner et al., 2020; Braconnot et al., 2021; Krätschmer et al., 2022), tuning of modelled dust emission rates against

observed dust deposition rates (Mahowald et al., 2006; Albani et al. 2014, 2015, 2016; Albani and Mahowald, 2019; Braconnot et al., 2021), and simulations with intermediate-complexity biogeochemistry models (Lambert et al., 2015, 2021; Heinemann et al., 2019). One key aspect missing from these compilations, except from the Holocene compilation by Albani et al. (2015), is a quantification of uncertainty in paleo-dust deposition rates. This is essential for compilations derived from multiple archive types, as each of these is subject to different sources of uncertainty. Explicit site-specific uncertainties in

paleo-dust deposition rates may be used for selecting subsets of observations against which to compare dust simulations, weighting observations for dust emission tuning purposes, or deriving distributions of global interpolations, for example through bootstrapping, Monte Carlo experiments, or Bayesian approaches.



During the seven years since the last paleo-dust compilation was published, there has been significant progress in the global coverage of paleo-dust proxies. Here, we update the previous compilations of global paleo-dust deposition flux and grain size observations with the newest data. Moreover, three improvements were introduced. First, given the expected high variability of dust deposition fluxes during the transition between the last glacial period and current interglacial (compared to that within any of those periods), we exclude any data corresponding to the last deglaciation to more precisely quantify dust deposition fluxes during the last glacial and current interglacial end-member climate states. Second, we derive site-specific, quantitative uncertainties of bulk and <10-µm paleo-dust deposition fluxes both for Holocene and last glacial observations. Third, we include for the first time dust deposition rate observations from peat bogs.

The manuscript is organized as follows: section 2 details our methodology in constructing Paleo±Dust, section 3 presents the main results, section 4 describes the structure of the datasets, and section 5 presents some concluding remarks.

## 2 Methods

### 2.1 Dust mass accumulation rate

The mean Dust Mass Accumulation Rate (*DMAR*) for a given time window at a location on the surface of the Earth may be quantified by dating two horizons along a vertical sedimentary profile ($t_{top}$, $t_{bottom}$), measuring the profile thickness between these two horizons ($h_{thick}$), its mean dry bulk density (*DBD*), and the mass fraction represented by atmospherically derived mineral dust not associated to direct volcanic ash fall (i.e., dust), henceforth defined as the eolian content (*EC*), as:

$$DMAR = \frac{h_{thick} * DBD * EC}{t_{bottom} - t_{top}}.$$

Given that bulk DMAR may be very sensitive to small, local dust sources, we also define a finer-grained DMAR, which is more representative of deposition of far-travelled dust derived from the main dust sources globally. A 10-µm dust diameter is used to define this finer-grained dust fraction, as this is either the uppermost or intermediate-bin threshold diameter that is found in the greatest number of CMIP6 Earth system models with dust representation (Zhao et al., 2022), thus facilitating model-data comparisons. It is also the recommended threshold definition between the fine-coarse and super coarse-giant size ranges of dust (Adebiyi et al., 2023). The fraction of DMAR that corresponds to dust particles less than 10 µm in diameter ($DMAR_{10}$) is simply:

$$DMAR_{10} = DMAR * f_{10},$$

where $f_{10}$ is the <10-µm dust mass fraction. This is the approach to calculating paleo-dust deposition fluxes used for all continental archives (i.e., loess-paleosol sequences, lakes, peat, and ice), and for a few sediment cores in the marine realm. Most DMAR constraints from marine sediment cores use the [230]Th normalization technique, which is for the most part independent of the profile chronology, as discussed further in section 2.8.

The absolute error on DMAR ($\varepsilon DMAR$) and $DMAR_{10}$ ($\varepsilon DMAR_{10}$) can be expressed as:



$$\varepsilon DMAR = DMAR \sqrt{\left(\frac{\varepsilon h_{thick}}{h_{thick}}\right)^2 + \left(\frac{\varepsilon DBD}{DBD}\right)^2 + \left(\frac{\varepsilon EC}{EC}\right)^2 + \frac{\varepsilon t_{bottom}^2 + \varepsilon t_{top}^2}{\left(t_{bottom} - t_{top}\right)^2}}$$

and

$$\varepsilon DMAR_{10} = DMAR_{10} \sqrt{\left(\frac{\varepsilon DMAR}{DMAR}\right)^2 + \left(\frac{\varepsilon f_{10}}{f_{10}}\right)^2},$$

where $\varepsilon h_{thick}$, $\varepsilon DBD$, $\varepsilon EC$, $\varepsilon t_{bottom}$, $\varepsilon t_{top}$ and $\varepsilon f_{10}$ are the absolute errors of $h_{thick}$, DBD, EC, $t_{bottom}$, $t_{top}$ and $f_{10}$, respectively. All errors reported in Paleo±Dust are Gaussian 1-σ uncertainties.

Our approach to assigning uncertainties to the components of DMAR combines objective and subjective considerations. On the one hand, when measurement uncertainties are reported in the original studies, these uncertainties are used. One example

is the use of [232]Th as a dust proxy to calculate EC in marine sediments. This method requires normalization of measured [232]Th in marine sediments to the mean global concentration of [232]Th in dust: the variability of this normalizing value provides a means to calculate the uncertainty in EC. Instead, when uncertainties are not reported, these are defined based on the distribution of reported relative uncertainties for sites of the same archive type (conservatively choosing the 75th percentile of this distribution). On the other hand, when the uncertainty for any component of DMAR is not reported, it is

assigned. Assigning uncertainty is more problematic as the true uncertainty is usually not known. This is the case for example for uncertainties in chronologies of loess-paleosol sequences derived through pedostratigraphy. In such cases, we assigned higher subjective uncertainties to methods deemed more uncertain (based on the authors' experience). This combination of approaches implies that reported DMAR uncertainties in Paleo±Dust should be considered to reflect the relative uncertainties among sites, more so than exact uncertainties for each site. Sections 2.3-2.8 expand on these

considerations for each archive type and DMAR component.

## 2.2 Criteria for inclusion of sites

Previous compilations of paleo-DMAR served as the starting point for the construction of Paleo±Dust. A literature search was performed to include new sites for studies published until (and including) December 2022. Recent advances in the study of peat bogs as a dust archive (e.g., De Vleeschouwer et al., 2014; Kylander et al., 2016, 2018; Marx et al., 2018; Sjöström et

al., 2020) have allowed the inclusion of this archive type for the first time in a dust compilation.

### 2.2.1 Time span

One of the main objectives of Paleo±Dust is to compile global constraints on DMAR for the last glacial-interglacial variability. In particular, Paleo±Dust compiles observations for the pre-industrial Holocene (pi-HOL), between year 1850 CE and 11.7 ka BP, and the Last Glacial Maximum (LGM), between 19.0 and 26.5 ka BP (Clark et al., 2009). We thus excluded

observations of DMAR for the present day and recent past, as well as for the last deglaciation (11.7-19.0 ka BP), to represent DMAR due to natural processes for the two main quasi-equilibrium climate states at glacial-interglacial timescales. Whether





this may be achieved or not for any particular site will depend on the sampling resolution along the vertical profile for dating. In order to include a site in Paleo±Dust, it needs to be possible to define a sub-section of the full sampled vertical section of a sedimentological archive whose time interval falls mostly (i.e., ≥75%) within either the pi-HOL or LGM as previously defined. Another criterion that needs to be fulfilled is that the site's time window is resolved within uncertainty: if $t_{bottom} \leq t_{top} + \varepsilon t_{top}$ and $t_{top} \geq t_{bottom} - \varepsilon t_{bottom}$, then that DMAR constraint is discarded.

Each entry in Paleo±Dust is associated to a specific time window within pi-HOL or LGM defined by $t_{bottom}$ and $t_{top}$. This allows for selecting sub-sets of data that include or exclude a specific time range of interest. As most of the previous compilations only report general time periods (e.g., Holocene, Late Holocene, LGM, last glacial), the original references of each entry in these compilations were revisited to extract exact values for $t_{bottom}$ and $t_{top}$. Given the imposed age criteria described above, most of the reported DMAR values in these compilations are different from those reported in Paleo±Dust for the same site, as a different set of samples from each site may have been used in this study to calculate DMAR.

### 2.2.2 Geomorphological setting

The most important consideration for selecting sites for Paleo±Dust is that the lithogenic fraction of the sediment be dominated by atmospheric deposition, or that the atmospherically derived lithogenic fraction may be estimated quantitatively. The site's geomorphological setting exerts a primary control on this. Sites of loess deposits with a massive structure reflective of deposition of dust from the atmosphere are included in the database, while *loessoid* sites consisting of reworked loess by either fluvial or slope processes are excluded (e.g., Pye, 1995). Peat usually consists of an upper ombrotrophic section where lithogenic materials are exclusively supplied by the atmosphere, and a lower minerotrophic section where lithogenic materials are also supplied laterally by groundwater (e.g., Shotyk, 1996). Only the ombrotrophic section of peat sections are considered in this study. In the case of lake sediments, non-eolian lithogenic material may be advected into a studied site through fluvial inlets, or via slope processes within the lake basin, a process known as lake sediment focusing (e.g., Blais and Kalff, 1995). We only included lake sites from endorheic basins, and discarded sites that lack a quantification of the fraction of lithogenic material advected through lake sediment focusing. With respect to marine sediments, we excluded sites located in continental margins, as they may be influenced by riverine inputs, as well as high-latitude sites potentially influenced by ice-rafted debris (e.g., Kienast et al., 2016). In both cases, exceptions are made to sites where these non-eolian lithogenic inputs are quantified. Other marine sites may be affected by sediment focusing, that is, lateral submarine transport of non-eolian sediment. The $^{230}$Th normalization method can be used to isolate the eolian lithogenic component in these cases (e.g., Francois et al., 2004). Finally, polar ice cores are retrieved from upland landscape positions, and for those sites that are located far from local lithogenic sources (those included in Paleo±Dust), their lithogenic materials are assumed to be wind-blown (i.e., Albani et al., 2015).

For loess, local geomorphology and topography can enhance dust deposition rate by focusing wind-blown material on the surface of windward escarpments (e.g., Comola et al., 2019), which may translate to local loess DMAR higher than the mean regional value (e.g., Xiong et al., 2015). This effect cannot be easily identified in the profile, particularly when a regional



loess stratigraphy is not available in the literature against which to compare the local stratigraphy. In general, upland loess sites are considered to be less prone to topography-induced enhancements in mass accumulation rates (Kohfeld and Harrison, 2003). We therefore performed a case-by-case evaluation of the geomorphological setting of each loess site, and discarded sites only when this potential problem is suggested explicitly in the original studies.

Conversely, the preservation potential of loess sites located in topographic lows within high-relief environments may be
limited due to slope or fluvial erosional processes that disrupt the continuity of the loess stratigraphy, generating hiatuses. These erosional hiatuses may be evident in the field or may only be discerned through high-resolution dating (e.g., Volvakh et al., 2022). For these reasons, we restricted the analyzed vertical extent of loess-paleosol profiles to sections with no intervening erosional hiatuses, be it that these can be directly inferred from the field, or that they are inferred from inversions in measured ages. We also excluded sites altogether that have a loess stratigraphy that considerably deviates from a well-
defined regional loess stratigraphy in regions where such a reference regional loess stratigraphy is available (based on the original authors' assessment).

### 2.3 Uncertainty in top and bottom age

Direct absolute dating of the sedimentary units of interest is the main tool for defining a chronology for dust records. The two most popular techniques are $^{14}$C and optically stimulated luminescence (OSL). When $t_{bottom}$ and $t_{top}$ are obtained by direct
absolute dating, the reported 1-σ uncertainties are used (Figure 1). If dating uncertainty is not reported, 3.4%, 9.1% and 13.1% relative uncertainties are assumed for $^{14}$C-, OSL- and thermoluminescence-based age determinations. These values represent the 75th percentile of the distributions of measured $^{14}$C (N = 83), OSL (N = 129), and thermoluminescence (N = 20) age relative uncertainties for sites included in Paleo±Dust. Absolute ages are usually reported as years before present, where the reference age (0 years BP) may be the year of sampling or a fixed year (e.g., 1950 CE). We do not homogenize
reported ages across studies so that they are referenced to the same year. Given that we focus on mean LGM and pi-HOL dust deposition rates, this does not introduce a significant uncertainty.

If a study presents a continuous age model, and this model includes, for example, the full LGM age interval (26.5-19.0 ka BP), then 26.5 ka BP is defined as $t_{bottom}$ and 19 ka BP as $t_{top}$ (Figure 1). If the study reports a modelled error to these interpolated ages, we used these reported values. If not, we calculated the errors in $t_{bottom}$ and $t_{top}$ as the L2 norm of the errors
of the two closest bracketing measured ages. The advantage of this error is that it is higher than the errors of either measured age. This extra uncertainty can be thought of as due to the interpolation.

When no absolute ages are available and the chronology is defined based on correlation of a sediment parameter with any given reference record, for example of the magnetic susceptibility signal in marine sediment cores or loess-paleosol profiles to the oxygen isotope record of marine benthic foraminifera (e.g., Lisiecki and Raymo, 2005), then a considerably higher
relative error is assigned (Kohfeld and Harrison, 2003; Figure 1). Wiers et al. (2019) quantified the uncertainty in the chronology of a Late Pleistocene, Arctic marine sediment core obtained by correlating magnetic properties in the core with global patterns of δ$^{18}$O in benthic foraminifera. They found a high absolute error during the last glacial-interglacial cycle of



approximately ±6 kyr, which remained relatively constant during this period. We apply this same absolute error for $t_{top}$ and/or $t_{bottom}$ obtained through such correlations (Figure 1). When these chronologies are in addition supported by absolute dating, we instead assumed a considerably lower absolute error of ±3 kyr. In the case of loess, chronologies may also be defined based on pedostratigraphy, by assuming that loess and paleosol units can be correlated to glacial and interglacial periods, respectively, as defined by marine isotope stratigraphy (Kohfeld and Harrison, 2003). Only pi-HOL dust deposition rates were obtained through this method (N = 11, all from China; Sun et al., 2000), as the LGM period cannot be discriminated from the encompassing last glacial period in terms of any pedostratigraphic unit. An evaluation of the uncertainty in the chronologies of these sites would require validation against absolute dates, which are not available for these sites. Moreover, to the best of our knowledge there are no similar validation studies for other sites that we could use as a model. We thus compensate this lack of validation by assigning a higher absolute uncertainty compared to magnetic correlations (±9 kyr).

For the specific case where the surface of a sedimentary unit is one of the bounding surfaces for a DMAR constraint and no continuous age model is available, $t_{top}$ is assumed to be 0 ka BP, and the uncertainty in this assumption is related to the difficulty in defining surface level, and is quantified in the following way (Figure 1): if for example $t_{bottom}$ is dated at 1.0 ± 10% ka BP at 15 cm depth, then in 1 cm centred at the top surface, the time span covered is 0.067 ka, assuming a linear relationship between age and depth. Then the age of the surface is fully contained within 0 ka BP ± (0.067/2) ka BP. Translating this absolute time interval to an equivalent 1-σ uncertainty interval, we multiply by 68.27%, and we get: 0 ka BP ± (0.067/2)*0.6827 ka BP. Finally, we multiply this 1-σ interval by 1.1, which corresponds to the relative age uncertainty for $t_{bottom}$ in our example (= 10%). The final surface age is 0.000 ka BP ± 0.025 ka BP (1σ). This procedure includes the uncertainty due to the difficulty in defining the surface level, and the uncertainty due to the dating method.

Finally, all polar ice core DMAR constraints in Paleo±Dust were obtained based on correlation to a common chronology for each of the polar regions: the 2012 Antarctic Ice Core Chronology (AICC2012, Veres et al., 2013) for Antarctic ice cores, and the 2005 Greenland Ice Core Chronology (GICC05, Svensson et al., 2008) for Greenland ice cores. In both cases, uncertainties in ages are as reported in each chronologic framework.

**2.4 Uncertainty in sediment profile thickness**

As $h_{thick} = h_{bottom} - h_{top}$, where $h_{bottom}$ and $h_{top}$ are the measured depths of the bottom and top layers, respectively, the error on the sedimentary thickness between dated layers is:

$$\varepsilon h_{thick} = \sqrt{\varepsilon h_{top}{}^2 + \varepsilon h_{bottom}{}^2} = \varepsilon h \sqrt{2},$$

where $\varepsilon h = \varepsilon h_{top} = \varepsilon h_{bottom}$ is the error on the measured depth to a dated layer beneath the surface level.

The uncertainty in measuring layer depth is associated to the fact that the sample obtained for dating has a finite vertical height, defined by the amount of sample mass required to perform dating. If, for example, loess sampling for dating is carried out by inserting horizontal corers with 2 cm in diameter into the vertical face of the sedimentary profile, then the full

sample will be included within ±1 cm from the centre of the corer. We may translate this total depth range to an equivalent range with 1-σ uncertainty by multiplying by 0.6827, resulting in an absolute 1-σ uncertainty in depth (εh) of ±0.6827 cm. If not reported, we assumed a horizontal corer diameter of 7.9 cm, which is the mean value reported for loess sites in Paleo±Dust. For lake, marine, peat and polar ice sampling where coring is performed vertically from the surface, the relevant quantity from which to calculate εh is the height of each sample (in the vertical direction along the corer) obtained after field

operations in the laboratory. If not reported, we assumed a sample height of 2 cm for marine sediment and peat bog records, and of 7.9 cm for lake sediment samples (in all cases the mean reported value for sample height in Paleo±Dust). All ice core sites in Paleo±Dust have reported sample height.

When the site chronology is based on magnetic susceptibility, εh is calculated as $εh = 0.6827 \times h_{MS}/2$, where $h_{MS}$ is the depth interval between susceptibility measurements.

### 2.5 Uncertainty in dry bulk density

In a study of forest soils in California, it was found that between three and 17 samples were needed to estimate mean soil DBD at ±10% at a 95% confidence level (Han et al., 2016). Here, a significantly higher DBD relative uncertainty of 15% (for 1σ) is assumed for sites with no reported DBD uncertainty, given that several sites have less than three DBD determinations, and a greater variety of sediment and soil types are considered in this compilation compared to the study by

Han et al. (2016). If, instead, the measurement error is reported in the original study, then this value is used. Finally, if DBD was measured but could only be estimated visually from a figure in the original study, a greater uncertainty of 20% is used. The same uncertainty is assigned when DBD was assumed equal to that of a near-by site where it was measured.

If DBD was not measured and the original study assumed a DBD value based on literature, an uncertainty of 30% is applied. If DBD was not measured and no specific value assumed in the original study, a value of 1.45 g cm$^{-3}$ and an uncertainty of

30% are used for loess and lake sites, except for loess sites in the Chinese Loess Plateau where a mean value of 1.48 g cm$^{-3}$ is preferred (Kohfeld and Harrison, 2003). For peat bog sites, when DBD is not measured (N = 5), the mean value of DBD among all sites with DBD measurements globally is used (= 0.12 g cm$^{-3}$, N = 18). No assumptions on DBD of marine sediment cores is required as DBD was always measured in sites that require DBD for the derivation of DMAR (those not based on $^{230}$Th normalization).

For polar ice cores, density is taken as that of ice: 916,750 g m$^{-3}$. This value is assumed to have a small variation between ±50 g m$^{-3}$, which translates to a 1-σ variability of ±50 g m$^{-3}$ x 0.6827 = 34.14 g m$^{-3}$. This approach is justified by the fact that density of ice after approximately 100 m depth below the surface almost constant with depth (e.g., Gerland et al., 1999).

### 2.6 Uncertainty in the mass fraction of dust

The calculation of the mass fraction of non-volcanic mineral dust (EC) and its uncertainty depends strongly on the type of

dust archive considered (Figure 2).

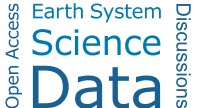

### 2.6.1 Loess-paleosol sequences

In loess studies with a focus on dust dynamics, EC is usually assumed to be 1, that is, loess is assumed to be fully composed of aeolian dust. However, while organic particles present in dust sources may be transported by wind and later deposited in the same manner as lithic particles (Muhs et al., 2014), here we assume that most organic matter in loess is post-depositional in origin (e.g., Hatté et al., 2001). Similarly, while carbonates constitute in some regions a significant fraction of airborne dust (Scheuvens and Kandler, 2014), we also assume that carbonates in loess-paleosol sequences are authigenic (e.g., Da et al., 2023). This is certainly not the case for some loess sites fed by carbonate-rich dust sources, as shown by Li et al. (2013) based on geochemical proxies. However, for most published studies it is not possible to calculate the contributions of primary and secondary carbonates to the total carbonate content of loess. Thus, when available, the organic matter and carbonate fractions are subtracted to derive EC. Relative 1-σ uncertainty in EC is assumed to be 10% (20%) if both (one of these two) fractions are (is) reported (Figure 2). If a correction for both fractions is available and there is also a quantification of volcanic inputs, then the relative uncertainty is reduced to 1%. Instead, when no such compositional information is available, loess-paleosol units are classified as either organic carbon-rich or organic carbon-poor, based on the physical description of the unit of interest, and EC is assigned a value of 0.94 and 0.98, respectively. These correspond respectively to the first and third quartiles of EC for sites where both the total organic matter and carbonate contents were determined (N = 28). In these cases, relative uncertainty in EC is highest at 30%.

### 2.6.2 Lake sediment cores

The procedure for calculating EC in lake sediment cores is similar to that for loess deposits, except that biogenic silica is an extra sediment component that can be very relevant in terms of mass and needs to be corrected for (Figure 2). Another effect to consider in lake sediment cores is sediment focusing, by which local sediments from the lake catchment can contribute to the siliciclastic mass. Some lake sediment studies isolate long-range dust from local catchment sediment, for example through grain size end-member modelling, trace element geochemistry (Petherick et al., 2009) or through corrections that consider lake bathymetry (Arcusa et al., 2020). Given the greater number of corrections required to calculate EC, the maximum potential relative uncertainty in EC is greater than for loess records (i.e., 50% versus 30%).

### 2.6.3 Marine sediment cores

A few studies of dust using marine sediment records calculate EC based on the same principles as with loess and lake sediment cores: by subtracting the carbonate, organic matter, and biogenic silica mass fractions, and when available, correcting for volcanic inputs (Figure 2). However, most studies that look at dust archived in marine sediment cores use isotope $^{232}$Th measurements to calculate EC, assuming a global mean concentration of $^{232}$Th in dust of 14.0 ± 4.6 ppm (Kienast et al., 2016; Ouyang et al., 2022). The relative uncertainty in this mean value (i.e., 1σ of 33%) is used as the



uncertainty in EC when calculated based on $^{232}$Th. Studies that used a different global mean value of $^{232}$Th concentration in dust were recalculated to this value.

When cores are extracted close to the continents, an important fraction of the siliciclastic mass may be due to riverine inputs. Following Singh et al. (2011) and Kienast et al. (2016), we excluded sites located less than 300 km from the coast, except for

the equatorial Atlantic Ocean off the coast of Brazil, for which this distance is 600 km (Holocene) and 800 km (LGM). This rule does not hold for studies that isolate dust from riverine contributions to terrigenous sediment (e.g., McGee et al., 2013).

Another potential terrigenous component that may obscure dust signals in marine sediment cores is ice-rafted debris. Here, we follow Kienast et al. (2016) and excluded marine sediment cores that are potentially affected by ice-rafted debris with high probability, that is, sites located poleward of 55ºN in the North Pacific, poleward of 50ºN in the North Atlantic, and

poleward of 50ºS in the southern oceans. These same latitudinal thresholds are applied to both pi-HOL and LGM paleo-dust sites. While the polar fronts may have changed latitudinally in time (and thus the extent of the influence of ice-rafted debris), there is no evidence of this from the relationship between latitude and paleo-lithogenic fluxes in marine sediment cores (Kienast et al., 2016). Sites located poleward of these latitudinal thresholds are kept when lithogenic fluxes are confirmed to be mostly due to eolian dust, for example through the measurement of n-alkanes, a proxy for continent-derived materials

(e.g., Lamy et al., 2014).

### 2.6.4 Polar ice cores

The EC fraction for the two Antarctic ice cores included in Paleo±Dust (both for the pi-HOL and LGM) was calculated based on Coulter counter insoluble particle volume concentration data, assuming an insoluble particle density of 2.5 g cm$^{-3}$ (Delmonte et al., 2005). The relative 1-σ uncertainty in Antarctic EC is 15.3%, which represents the L2 norm of a 11.4%

error component due to uncertainty in insoluble particle density (2.0-2.8 g cm$^{-3}$, Kohfeld and Harrison, 2001), a 10% error component due to the lack of a volcanic input correction, and a 2% component derived from replicate Coulter counter measurements (Figure 2).

In the case of Greenland ice cores, EC is based on $\delta^{18}$O and Ca$^{2+}$ concentration data. Sample depths with $\delta^{18}$O < -40‰ (cold periods) are assigned a Ca-to-dust ratio of 0.095, while sample depths with $\delta^{18}$O > -37‰ are assigned a Ca-to-dust ratio of

0.26 (Steffensen, 1997; Ruth et al., 2002), with a linear interpolation for in-between values of $\delta^{18}$O (Albani et al., 2015). Uncertainty in EC is assumed to be 22.4%, which is the L2 norm of a 10% uncertainty component due to possible volcanic inputs (as for Antarctic cores) and 20% due to a combination of analytical and Ca proxy uncertainties (Figure 2; Albani et al., 2015).

### 2.6.5 Peat cores

There are multiple ways in the literature to calculate EC in peat bogs. One way is by measuring the fraction of non-combustible mineral ash in total dry peat mass, which we can approximate as the siliciclastic mass fraction (e.g., Martínez Cortizas et al., 2020). A preferred approach is to use the concentration of one (e.g., Sharifi et al., 2018) or multiple (e.g.,



Pratte et al., 2020) conservative lithogenic elements that are not subject to post-depositional mobilization, and that have no anthropogenic source, such as Sc, Zr, Y and the REEs, normalized typically to the mean concentration of these elements in

the upper continental crust. Irrespective of the method employed, the lowest relative uncertainty in the calculation of EC is assigned to cases where the choice of the dust proxy or combination of dust proxies is based on a principal component analysis (PCA, Figure 2). This is because EC is sensitive to the choice of the geochemical proxy for dust (Shotyk et al., 2002; E. Resongles, pers. comm.), and a PCA analysis usually provides the best way to identify the element or set of elements that best represents atmospherically derived lithogenic fluxes. However, a high variability in DMAR calculated

with different elements that were all found to be associated with dust based on PCA analyses has also been reported (Kylander et al., 2016). This stresses the need to carefully select the elements to be used for DMAR calculations based on the site-specific element behaviour. Specifically, it is recommended that when more than one element is identified as the optimal dust proxy, that DMAR be calculated based on the combination of those elements (e.g., based on the sum of REEs, Sc, Y and Zr if all these elements are identified as equally suitable to represent dust).

To be consistent with the choice of uncertainties in EC for loess-paleosol sequences, lakes and marine sediments, a relative uncertainty of 10% is assigned to PCA-supported EC calculations for peat bogs (Figure 2). If no PCA analysis is available, DMAR calculated based on multiple elemental proxies is preferred over a single-proxy approach, including the case when that single proxy is the non-combustible ash fraction. Moreover, when independent information is available that allows identifying samples with high inputs of direct volcanic ash fall, such as based on Nd isotopes (e.g., Vanneste et al., 2015), a

correction is applied by which high DMAR values for those samples are replaced by background DMAR values in the same profile.

### 2.7 Uncertainty in the <10-µm dust mass fraction

We assign the lowest uncertainty to $f_{10}$ when it is calculated from a full grain size distribution, derived either from particle counter or laser diffraction measurements (Figure 3). This uncertainty is associated with the reproducibility of measurements

(~0.7% after propagation of each bin's uncertainty, V. Nogués, pers. comm.) and for all archive types except ice cores also with the efficiency of manual or mechanical sieving (e.g., ~14% for a 62.5-µm sieve, Lopez-García et al., 2021), among other sources of error. Here, we assign a conservative 20% relative uncertainty to $f_{10}$ based on these methods for all archive types except for ice cores, which are assigned a relative uncertainty of 1%. In many cases where such measurements are carried out, the full-size distribution data is not published and $f_{10}$ is not reported in the original study. For a number of

Holocene sites published previously to 2015, Albani et al. (2015) compiled these distributions. For these sites, $f_{10}$ was retrieved from that study. For the rest of the sites in Paleo±Dust, if data was not available but the grain size distribution was plotted in the original study, we estimated $f_{10}$ visually adding an extra 10% of uncertainty (30% in total). To perform this visual estimation, we used a vector graphics editor (Adobe Illustrator) to draw two polygons: one that encompassed the area under the curve of the volumetric grain size distribution for all measured particle sizes, and one for sizes <10 µm. We

calculated $f_{10}$ as the ratio of the latter to the former area. Greater uncertainty is associated to $f_{10}$ when it is calculated from





grain size bins by using the sieve and pipette method (30%), or when calculated from a reported mean value (40%), based on assumptions of the underlying grain size distribution. When no grain size measurements are available for a given site, but only for near-by sites that are comparable in terms of their geomorphological setting, then the same value for $f_{10}$ is used in both sites, with an extra 10% uncertainty for the site with no data. The same is true for sites that include grain size data for a

different time window than the one considered.

For marine sediment cores, it is rarely the case that grain size information is available (only 6% of sites). For sites with no grain size information, if they are within ±5 degrees in latitude and longitude from a site with grain size measurements and provided the two sites have similar bathymetries and are not separated by significant bathymetric features, then the same $f_{10}$ value is used for both sites, with an extra 10% uncertainty for the site with no measurements. For all other marine sediment

sites, $f_{10}$ values are assumed based on their downwind distance from known dust sources. If sites are located >2000 (1000-2000, <1000) km downwind from known dust sources south of 15ºS, we assume $f_{10}$ = 1.00 (0.75, 0.50) ± 60%. For locations in the ocean north of 15ºS where dust sources are more intense, these threshold distances are instead 3000 and 1500 km. The downwind directions from dust sources were qualitatively defined based on Holocene and LGM dust deposition maps from Mahowald et al. (2006), although dust sources not considered in this study such as Alaska were also considered.

**2.8 The use of 230-thorium normalization for total sediment flux calculations in marine sediment cores**

It has long been recognized that the traditional method of obtaining lithogenic mass accumulation rates between two dated horizons in marine sediment cores (e.g., Mortlock et al., 1991) cannot be directly attributed to pelagic sedimentation from the water column due to the process of sediment focusing, which is the lateral transport of sediment acting on the seafloor and along different depths in the ocean. Instead, the majority of lithogenic flux estimations from marine sediment cores

interested in the vertical fluxes of particles use the $^{230}$Th normalization method, which accounts for sediment focusing and provides estimations of pelagic mass accumulation rates. This method was first proposed by Bacon (1984) and is fully described in Francois et al. (2004). Briefly, the main assumption of the method is that the flux of scavenged $^{230}$Th to the seafloor is equal to the decay production rate of $^{230}$Th from $^{234}$U dissolved in the overlying water column. This is a reasonable assumption given the short residence time of $^{230}$Th in ocean water, and the fact that its removal from the dissolved

pool is mostly through adsorption to sediments derived vertically in the water column, a process known as proximal scavenging.

The DMAR calculation for the case of marine sediment cores using the $^{230}$Th normalization technique is calculated as the product of the sediment bulk mass accumulation rate (SBMAR) and EC:

$DMAR = SBMAR * EC,$

where

$$SBMAR = \frac{\beta_{230} * z}{Th\_230^{°}_{xs}} \text{ (e.g., Francois et al., 2004).} \quad \quad \text{(eq. 1)}$$



Here, $\beta_{230}$ is the decay constant of [234]U (and production constant of [230]Th) throughout the water column, with a value of 2.562 x 10[-5] dmp cm[-3] kyr[-1] (Costa et al., 2020), $z$ is the water depth to a given site, and $Th\_230^{°}_{xs}$ is the decay-corrected excess [230]Th activity (in dmp g[-1]).

Another advantage of the [230]Th normalization method is that it provides DMAR estimates that are for the most part independent of the chronology of sediment accumulation. This is particularly relevant for high-resolution DMAR studies as large uncertainties exist for DMAR using the traditional method when estimates are attempted for two horizons located close to each other in age.

The relative uncertainty in deriving SBMAR based on the [230]Th normalization method is 30% (Figure 4), based on 390 calibration studies (Henderson et al., 1999; Scholten et al., 2001; Yu et al., 2001a, 2001b).

**2.8.1 Extra uncertainty in DMAR due to downslope sediment flow**

If particles settling into the studied site directly from the overlying water column have the same [230]Th activity as particles that have previously been advected laterally for some distance, then eq. 1 can be used as discussed before to calculate pelagic SBMAR. This is typically the case when laterally advected sediment is resuspended by bottom currents from a position in 395 the seafloor at a similar water depth compared to the studied site (Francois et al., 2004). However, when the [230]Th activity of both sediment components differ, inaccuracies may appear in the calculation of pelagic SBMAR. In principle, this may be the case when laterally advected sediment is originated at positions in the seafloor shallower than the studied site (note dependence of SBMAR on the water depth in eq. 1) and transported by either downslope bottom currents or by intermediate nepheloid currents (Francois et al., 2004). This is favoured in regions of the seafloor with high regional bathymetric 400 gradients, such as along the continental slope or in the open ocean close to bathymetric features (e.g., seamounts, aseismic ridges).

Only a few studies carried out bathymetric analyses to derive explicit corrections to their SBMAR estimates to account for this potential effect of downslope sediment flow. One such example is from a site located at the foot of the Sierra Leone Rise in the equatorial Atlantic Ocean (EN066-29GGC, Francois et al., 1990). Based on a detailed analysis of the surrounding 405 bathymetry, these authors defined a worst-case scenario in which all laterally transported sediment originated from the topmost part of the rise (with the greatest possible difference in water depth with the studied site and thus the greatest deviation in [230]Th activity). In this scenario, the true pelagic SBMAR was overestimated by 36.5% compared to SBMAR as calculated with eq. 1 (Francois et al., 1990). Nonetheless, the true overestimation was probably lower as a more realistic scenario is one in which laterally transported sediment was not fully derived from the top of the rise, but rather partially from 410 different steps along the rise at different water depths (Francois et al., 1990).

Instead, a study that looked at six cores in the Juan de Fuca Ridge separated <50 km from each other found no systematic differences in [230]Th-normalized sediment fluxes, despite hundred-metre-scale relief between sites (Costa and McManus, 2017). Whether this is an indication of the lack of sensitivity of [230]Th-normalized sediment fluxes to downslope sediment flow at a global level remains to be determined. Because of this uncertainty, here we do not attempt to apply corrections to





$^{230}$Th-normalized DMAR estimates from marine sediment cores to account for this effect. Another reason for not doing so is that it would require an in-depth, site-specific bathymetric analysis, which is out of the scope of this study. Instead, we raised the uncertainty in SBMAR for those sites that have probably experienced sediment focusing through downslope flow. We did so for sites that (i) are located in positions in the seafloor with high regional bathymetric gradients, (ii) are located at a relatively deep position compared to its surroundings, and (iii) have experienced sediment focusing. We evaluated the first

two criteria by using GEBCO_2022, a global bathymetric grid at 15 arc-second horizontal resolution (GEBCO Compilation Group, 2022). For each marine sediment core site in Paleo±Dust, we took a 5-degree latitude by 5-degree longitude area centred at the site and calculated the difference between the 5th and 95th percentiles in bathymetry for all GEBCO_2022 grid cells located in that area. We also calculated this value for a reference marine sediment core site: EN066-29GGC (Francois et al., 1990). Any given marine sediment core site in Paleo±Dust where this 5th-95th percentile difference was

equal to or higher than half of the 5th-95th difference for the reference site was considered to satisfy condition (i). Condition (ii) was satisfied for those sites located at a water depth of at least the median of that of the area defined for each site. Finally, condition (iii) was satisfied for those sites with a sediment focusing factor greater than one, where the focusing factor is the ratio of laterally advected to vertical sediment flux, as defined by Francois et al. (2004) and retrieved for each site either from the original study or from the compilation by Costa et al. (2020).

Those sites that passed the above criteria were assigned an extra uncertainty component to SBMAR, combined based on the L2 norm to the base relative uncertainty component of 30% common to all SBMAR estimates for marine sediment cores arising from $^{230}$Th normalization (Figure 4). This extra component is calculated for each site based on a realistic scenario for reference site EN066-29GGC (half of the worst-case scenario, Francois et al., 1990), and is proportional to the site's focusing factor.

**3 Results**

Paleo±Dust consists of a total of 284 pi-HOL and 208 LGM sites, of which approximately a third are sites published since 2016 and not included in previous paleo-dust deposition flux compilations (Table 1). Of all sites, 52% correspond to loess, 39% to marine sediment cores, 5% to peat cores, 2% to lake cores and 2% to polar ice cores. All peat sites are new to this compilation, with 65% of the sites published since 2016. Loess is the archive type with the highest number of new sites

published since 2016 at 89, and except for peat sites, it is also the archive type with the highest percentage of new sites at 35%. All DMAR determinations from peat sites are for pi-HOL, which can be explained by the fact that the vast majority of peat bogs globally only started to form during the last deglaciation (Yu et al., 2010).

While there are several studies that report dust measurements in non-polar ice caps and mountain glaciers for the pi-HOL and LGM, the reported quantity is in all cases the particle number concentration (Fisher, 1979; Thompson et al., 1989, 1995,

1997, 1998; Clifford et al., 2019; Beaudon et al., 2022), except for a site on the Penny ice cap where the mass concentration is reported (Zdanowicz et al., 2000). Many of these studies also lack grain size data. The derivation of mass deposition rate from particle number concentration alone is not straightforward and requires several assumptions (Kohfeld and Harrison,





2001), including the grain size distribution and the distribution of particle density with grain size. We thus follow Kohfeld
and Harrison (2001) and do not derive dust deposition rates in these cases. Thus, no ice cores from non-polar sites are

included in Paleo±Dust.

There is a clear Southern versus Northern Hemisphere asymmetry in the number of sites in Paleo±Dust (Figure 5a-b). This is
mostly evident in the number of continental dust archives. Only a few loess studies exist from the Pampean region in South
America, the main loess belt in the Southern Hemisphere, from which DMAR estimates can be obtained (Kemp et al., 2004;
Torre et al., 2019; Coppo et al., 2022). South American loess is restricted to latitudes <40ºS, and thus it only archives dust

from low to mid-latitude sources in South America, as well as from northernmost Patagonia. Emissions from southernmost
Patagonia's main dust sources during the pi-HOL are captured by peat bogs on Malvinas Islands (Monteath et al., 2022), and
possibly as well by peat bogs located to the east of the Andes (Vanneste et al., 2015, 2016). These latter studies report on
sites that are located upwind of the main present-day dust sources, but that have captured pi-HOL dust from smaller sources
in the region. Instead, the bulk of dust emissions from central and southern Patagonia (40-50ºS) since the LGM are largely

unconstrained due to a lack of loess deposits downwind. It is thus critical that marine sediment cores are drilled in the open
ocean off the eastern Patagonian coast with a focus on constraining dust fluxes. Elsewhere in the Southern Hemisphere, there
is a complete dearth of DMAR estimates from loess or loess-like deposits in Australia, and only one pi-HOL DMAR
estimate in southern Africa (Brunotte et al., 2009). In the Northern Hemisphere, the high latitudes in Asia (i.e., Siberia)
constitute the area with the scarcest coverage of DMAR constraints. In turn, Europe is the region with the highest number of

new DMAR estimates since 2016.

Based on 125 sites with paired LGM and pi-HOL DMAR determinations (Figure 5c-d), the global LGM:pi-HOL ratio of
DMAR is $3.4 \pm 0.7$, and of $DMAR_{10}$ is $3.1 \pm 0.8$ ($1\sigma$, Table 1). This latter value is equivalent within error to that determined
exclusively based on loess $DMAR_{10}$ determinations ($3.2 \pm 0.8$, N = 47). Based exclusively on intermediate-range dust
deposition as archived in marine sediments, this ratio is significantly lower, albeit within the same order of magnitude ($2.2 \pm$

$0.7$, N = 72). Finally, based only on dust archived in polar ice cores, this ratio goes up dramatically to $16.8 \pm 0.7$, although
this is based only on four sites. Three processes are responsible for higher LGM/pi-HOL dust deposition rate ratios for polar
ice cores compared to loess: (1) the effect of dust transport in the atmosphere and dry deposition, by which small changes in
dust emission intensity translate to small changes in close-to-source dust deposition and to comparatively bigger changes in
remote dust deposition (Lambert et al., 2008), (2) the effect of mid-latitude precipitation on wet scavenging of dust en route

to the poles (Markle et al., 2018), whose variability at glacial-interglacial timescales may imply an amplification of dust
deposition rate variability at the poles compared to the low latitudes, and (3) specifically for the Southern Hemisphere, the
activation or intensification of dust emissions from Patagonian sources at higher latitudes (compared to the present day)
during the LGM implied more efficient transport of dust to Antarctica (Andersen et al., 1998; Petit et al., 1999; Lambert et
al., 2008; Albani et al., 2012).

Relative $1$-$\sigma$ uncertainties in DMAR and $DMAR_{10}$ mostly range between 21-98% and 37-106%, respectively (5%-95%
percentiles, Figure 5e-f). The least uncertain archive type for DMAR and $DMAR_{10}$ is polar ice (median: 23% and 24%,





respectively, N = 8), while the most uncertain are marine sediments (44% and 74%, N = 191). However, the differences in median relative uncertainties among archive types is significantly lower than the differences among individual sites across archive types. There are no significant differences in the uncertainty of LGM vs. pi-HOL DMAR and DMAR$_{10}$. The

DMAR$_{10}$ determinations with the greatest uncertainties (the top 5% percentile, with values >114%) are all in loess deposits that either lack several determinations and so many assumptions are made (e.g., <10-µm grain size fraction, density, organic content, carbonate content) and/or have $t_{top}$ and $t_{bottom}$ values that are distinct within error, yet very close between them.

## 4 Structure of Paleo±Dust

All Paleo±Dust files are included in the supplementary information file *supplementary_data.zip*. It consists of two main tab-

delimited text files containing the most important variables for each site (i.e., main_piHOL_tab.txt, main_LGM_tab.txt), two extra supporting tab-delimited text files with an expanded set of intermediate variables used to calculate the main set of variables (i.e., supporting_piHOL_tab.txt, supporting_LGM_tab.txt) and two separate text files (i.e., site_specific_notes_references.txt, site_references.txt) containing site-specific observations and a list of references from where the necessary data was extracted.

The two main text files each contain 10 variables: *locality* ('siteName_region'), *type* (i.e., 'ice-core', 'marine', 'loess', 'lake' or 'peat'), *lat_N* (i.e., latitude in degrees north with two decimal places, between -90 and 90), *lon_E* (i.e., longitude in degrees east with two decimal places, between -180 and 180), *top-age_kaBP* and *bottom-age_kaBP* (i.e., $t_{top}$ and $t_{bottom}$, respectively, in thousands of years before present, with two decimal places), *DMAR_g/m2/a* and *DMAR-1sigma_g/m2/a* (i.e., mean DMAR and 1-σ εDMAR for the defined age bracket, respectively, in g m$^{-2}$ a$^{-1}$, with three significant digits), and

*DMAR10_g/m2/a* and *DMAR10-1sigma_g/m2/a* (i.e., mean DMAR$_{10}$ and 1-σ εDMAR$_{10}$ for the defined age bracket, respectively, in g m$^{-2}$ a$^{-1}$, with three significant digits).

The two supporting text files each contain 24 variables, corresponding to the same 10 variables as the main files plus 14 extra variables: *first_appeared_in_dataset* (i.e., reference to compilation where site first appeared), *top-age-1sigma_kaBP* and *bottom-age-1sigma_kaBP* (i.e., 1-σ ε$t_{top}$ and ε$t_{bottom}$, respectively, in thousands of years before present, with two decimal

places), *thickness_m* (i.e., $h_{thick}$ defined by the age bracket, in m, with two decimal places), *depth-1sigma_cm* (i.e., 1-σ uncertainty of the depth below the surface of the dated layers, in cm, with two decimal places), *DBD_g/m3* and *DBD-1sigma_g/m3* (i.e., DBD and 1-σ εDBD, respectively, in g m$^{-3}$, with no decimal places), *SBMAR_g/m2/a* and *SBMAR-1sigma_g/m2/a* (i.e., SBMAR and its 1-σ uncertainty, respectively, only for type 'marine', in g m$^{-2}$ a$^{-1}$, with three decimal places), *EC_adim* and *EC-1sigma_adim* (i.e., EC, as a number between 0 and 1, and 1-σ εEC, respectively, both with three

significant digits), *f10_adim* and *f10-1sigma_adim* (i.e., $f_{10}$, as a number between 0 and 1, and 1-σ ε$f_{10}$, respectively, both with three decimal places) and *flag-marine-sed-downslope* (i.e., a flag exclusive to type 'marine' where 1 means that the site is prone to contamination by downslope sediment movement, and where 0 means the site is not prone to such contamination).

## 5 Conclusions

Paleo±Dust is an updated global paleo-dust deposition rate compilation for mean pi-HOL and LGM climate states that includes quantitative estimates of uncertainties. Paleo-dust flux measurements from peat bog cores are included in a paleo-dust compilation for the first time. By excluding deglaciation dust fluxes, Paleo±Dust better isolates mean interglacial and glacial dust fluxes than previous datasets. Site-specific age brackets allow to sub-sample mean dust deposition constraints for specific time windows of interest. Grain size information is extracted from the original studies to derive fluxes of dust with

<10-µm diameter particles.

The main feature of Paleo±Dust is the inclusion of site-specific dust flux uncertainties that are consistent across paleo-dust archive types, that is, that can be used to gauge the relative accuracy in dust flux constraints between sites of different geologic nature. These new uncertainty data may also be of use as a criterion for selecting a sub-set of samples for comparison against dust simulation output, to tune dust emission in Earth system models using dust deposition flux data and

a weighted approach based on uncertainty of proxy data, or to construct distribution-based global interpolation maps of paleo-dust deposition rates through Bayesian approaches, Monte Carlo, or bootstrapping experiments for use for example as input in biogeochemical models.

## Data availability

For review purposes, the datasets are available as part of the supplementary information of this submission (file

supplementary_data.zip). Paleo±Dust has been uploaded to the Pangaea public repository (https://www.pangaea.de/) under a CC-BY license, and we are currently awaiting a DOI from Pangaea, which will be available once the review process is through.

## Acknowledgements

We thank D. Muhs, D. Constantin, T. Stevens, S. Pratte, and G. Le Roux for providing us with highly valuable data. This

research was financed by project ANID-FONDECYT-POSTDOCTORADO2020-3200085 and by a 2022 INQUA Fellowship, both awarded to NJC, and by project ANID-FONDECYT-REGULAR2023-1231682 awarded to FL. The discussions that resulted in this paper also benefited from meeting travel grants from project CLIMAT-AmSud 22-CLIMAT-01.

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





**Loess-paleosol Lake Peat Marine**

① Discrete absolute dating ─┬ *reported 1σ*

└ *if not reported, 3.4% ($^{14}$C), 9.1% (OSL), 13.1% (TL) relative 1σ*

❗ Surface age = 0 ka BP: uncertainty due to bottom age analytics and difficulty in defining surface level

② Indirect dating based on discrete absolute ages ─┬ *topmost (bottommost) of target under(over)lying stratigraphic unit: 10% relative 1σ*

├ *general mean (bulk) age of under(over)lying unit: 20% relative 1σ*

└ *assumed equal to regional dated markers: 30% relative 1σ*

③ Continuous age model (e.g., polynomial, bayesian) ─┬ *reported 1σ uncertainty envelope*

└ *if not reported, uncertainty is the quadrature of the uncertainties of bracketing dated levels*

④ Correlation (MS, δ$^{18}$O) ─┬ *not supported by absolute dating: ±6 kyr absolute 1σ*

└ *supported by absolute dating: ±3 kyr absolute 1σ*

⑤ Correlation (pedostratigraphy, applies only to loess) ── *correlation to regionally defined loess-paleosol chronology: ±9 kyr absolute 1σ*

**Polar ice**

Correlation (AICC2012, GICC05) ── *polar ice cores are correlated to the AICC2012 and GICC05 chronologies for Antarctic and Greenland cores, respectively: uncertainty as reported in these reference chronologies*

**Figure 1: Criteria for assigning uncertainty to the top and bottom ages. OSL: optically stimulated luminescence, TL: thermoluminescence, AICC2012: Antarctic Ice Core Chronology 2012 (Veres et al., 2013), GICC05: Greenland Ice Core Chronology 2005 (Svensson et al., 2008).**




**Loess-paleosol**

① Correction for carbonates, OM, volcanic inputs — *all three (two, one): 1% (10%, 20%) relative 1σ*

② Assumed based on physical description: loess (= 0.98) vs. (paleo)soil (= 0.94) — *30% relative 1σ*

**Lake**

① Correction for carbonates, OM, bSiO₂, volcanic inputs, sediment focusing — *all five (four, three, two, one): 1% (10%, 20%, 30%, 40%) relative 1σ*

② Assumed — *50% relative 1σ*

**Marine**

① Correction for carbonates, OM, bSiO₂, river and volcanic inputs — *all five (four, three, two, one): 1% (10%, 20%, 30%, 40%) relative 1σ*

② Based on ²³²Th — *33% relative 1σ*

**Polar ice**

① Antarctic ice cores: based on Coulter counter insoluble particle volume concentration data (no volcanic correction) — *15.3% relative 1σ*

② Greenland ice cores: based on assumed δ¹⁸O vs. calcium:dust concentration ratio (plus no volcanic correction) — *22.4% relative 1σ*

**Peat**

① PCA guides decision on dust geochemical proxies — *10% relative 1σ*

  ↳ + volcanic correction applied (e.g., Nd isotopes) — *1% relative 1σ*

② No PCA, multi-proxy approach, volcanic correction — *20% relative 1σ*

③ No PCA nor volcanic correction, multiple dust proxies considered — *30% relative 1σ*

④ No PCA, single-proxy approach, volcanic correction — *50% relative 1σ*

⑤ No PCA, single-proxy approach, no volcanic correction — *60% relative 1σ*

**Figure 2: Criteria for assigning uncertainty to the non-volcanic dust fraction. OM: organic matter, bSiO₂: biogenic silica, PCA: principal component analysis.**





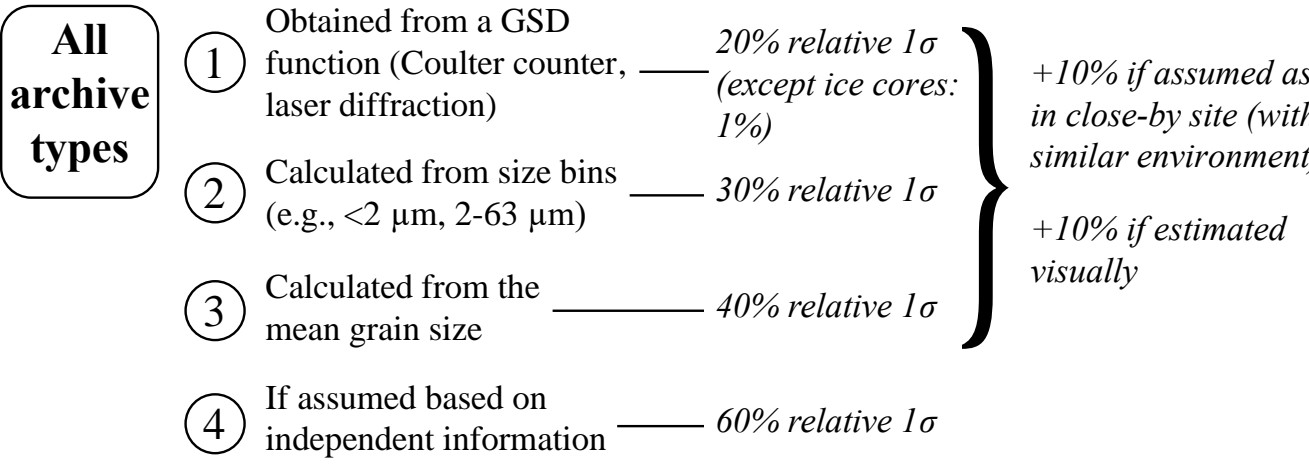


**Figure 3: Criteria for assigning uncertainty to the <10-µm grain size fraction. GSD: grain size distribution.**

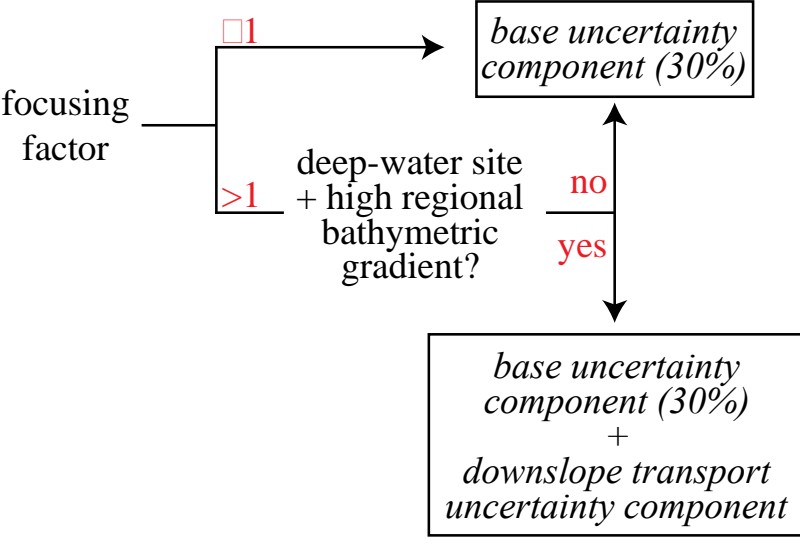

**Figure 4: Criteria for assigning uncertainty to the sediment bulk mass accumulation rate when using the $^{230}$Th normalization technique for deriving dust fluxes from marine sediment cores.**




**Figure 5: Distribution of sites for the (a) pre-industrial Holocene (pi-HOL) and (b) Last Glacial Maximum (LGM) time periods, (c)**
**pi-HOL and (d) LGM <10-µm dust deposition rate, and (e) pi-HOL and (f) LGM <10-µm dust deposition rate uncertainty.. The**
**coastlines for all panels correspond to those of the present day.**





**Table 1: Number of sites in Paleo±Dust for the pre-industrial Holocene (pi-HOL, year 1850 CE-11.7 ka BP) and Last Glacial Maximum (LGM, 19.0-26.5 ka BP) with a dust deposition rate determination. For the mean LGM/pi-HOL dust flux ratio (<10-μm fraction) calculations, only sites with both pi-HOL and LGM determinations were considered.**

| | # of pi-HOL sites (published since 2016) | # of LGM sites (published since 2016) | LGM/pi-HOL dust flux ratio (<10-μm fraction, ± 1$\sigma$) |
|---|---|---|---|
| Polar ice | 4 (0) | 4 (0) | 16.8 ± 0.7 (N = 4) |
| Marine sediments | 93 (18) | 98 (29) | 2.2 ± 0.7 (N = 72) |
| Loess | 154 (51) | 104 (38) | 3.2 ± 0.8 (N = 47) |
| Peat | 23 (15) | 0 (0) | - |
| Lakes | 10 (4) | 2 (0) | 1.9 ± 2.8 (N = 2) |
| TOTAL | 284 (92) | 208 (63) | 3.1 ± 0.8 (N = 125) |