# Peer review of "Paleo±Dust: Quantifying uncertainty in paleo-dust deposition across archive types"

_Earth System Science Data, 2023_

## Referee Comment (RC1)

**Review of „Paleo±Dust: Quantifying uncertainty in paleo-dust deposition across archive types" Cosentino et al.**

**Manuscript number: ESSD-2023-241**

In recent years, there has been a renaissance in the field of spatio-temporal reconstruction of sediment mass accumulation rates (MAR), probably due to the fact that atmospheric dust does indeed play a significant role in shaping Earth's climate, in both direct and indirect ways. Yet there are considerable uncertainties about the actual role of dust that need to be clarified in order to better understand how the Earth system works under present-day or past (e.g. glacial) climate conditions. Research on MAR reconstructions is taking several directions: 1) synthesis of new data from one or more type(s) of paleo-dust archives, 2) methodological innovations in the calculation of MAR, 3) database construction, and/or 4) focusing on periods that have not been included in reconstructions so far (e.g. Marine Isotope Stage 4-5-6). As such, a new database called ChronoLoess (Bosq et al.) is currently appearing in the ESSD pages, which can be extended in a standardised format. In contrast to ChronoLoess, which focuses on only one archive, the authors of this manuscript have compiled data from a large number of paleodust archives of different types (marine/terrestrial) in the Paleo±Dust database, following a common methodology. I consider the most important advance of the present paper to be that they have discussed the uncertainties associated with the dust mass accumulation rate (DMAR) calculations of each archive type and attempted to estimate them quantitatively, which in many cases is not a simple task. In addition, paleo-dust flux estimates from peat bog cores are included in the paleodust compilation for the first time.
Having read the manuscript, I find it to be neatly written, logical and easy to follow, and I consider it to be a significant step forward in the reconstruction and comparison of LGM and Holocene dust fluxes. The authors have tried to organise the database in a way that it can be used for future Earth system modelling, which is also a positive aspect. I have, of course, some critical comments on the manuscript, which I write about below (and in the annotated pdf file attached), but these do not affect the conclusions to any significant extent and are rather just a further refinement of the methodology. On the whole, the manuscript, with some modifications, can be published in ESSD.

Line-by-line comments

Line 97, εDMAR: This is just a pedantic point, but the error of a quantity is usually denoted by the Greek sigma in statistics, I have never seen the use of epsilon for that purpose. What is the reason for using epsilon here? There is another reason of why I discourage the authors using epsilon: in dust provenance studies the Nd and Hf isotope compositions are often expressed in the epsilon notation (εNd, εHf), so this may potentially cause some misunderstandings.

Lines 98-100: The authors presented uncertainty estimations for MAR and MAR10, which disregard potential correlations between the variables in the MAR equations, so these are not full statistical treatments. I suggest the error propagations below.

Let us rewrite the DMAR equation: $DMAR = \frac{h_{thick} \times DBD \times EC}{t_{bottom} - t_{top}} = \frac{h_{thick} \times DBD \times EC}{\Delta t} = \frac{x \times y \times z}{\epsilon}$ ($\epsilon = \Delta t = t_{bottom} - t_{top} = t_b - t_t$)

The errors (or variances, or standard deviations) of x, y, z and $\epsilon$ ($\sigma_x^2 = s_{h_{thick}}^2$, $\sigma_y^2 = s_{DBD}^2$, $\sigma_x^2 = s_{EC}^2$, $\sigma_\epsilon^2 = s_{\Delta t}^2$) are known. Obviously, x ($h_{thick}$)and $\epsilon$ ($\Delta t = t_{bottom} - t_{top}$) are correlated ($C_{x\epsilon} \neq 0$, $C_{x\epsilon}$ is the covariance of x and $\epsilon$, and $C_{x\epsilon} = \varrho_{x\epsilon}\sigma_x\sigma_\epsilon$, where $\varrho_{x\epsilon}$ is correlation of x and $\epsilon$), while no correlation exists between the other variables: $C_{xy} = C_{yz} = C_{y\epsilon} = C_{z\epsilon} = 0$. DBD and EC may be correlated, this can be a matter of debate, but we suppose zero correlation here. With this, the uncertainty of DMAR is given by:

$$\sigma_{DMAR}^2 = f^* \times M \times f$$

, with $f^* = \left(\frac{\partial DMAR}{\partial x}, \frac{\partial DMAR}{\partial y}, \frac{\partial DMAR}{\partial z}, \frac{\partial DMAR}{\partial \epsilon}\right)$, $f = \left(\frac{yz}{\epsilon}, \frac{xz}{\epsilon}, \frac{xy}{\epsilon}, -\frac{xyz}{\epsilon^2}\right)$, $M = \begin{bmatrix} \sigma_x^2 & C_{xy} & C_{xz} & C_{x\epsilon} \\ C_{yx} & \sigma_y^2 & C_{yz} & C_{y\epsilon} \\ C_{zx} & C_{zy} & \sigma_z^2 & C_{z\epsilon} \\ C_{\epsilon x} & C_{\epsilon y} & C_{\epsilon z} & \sigma_\epsilon^2 \end{bmatrix}$

$$\sigma_{DMAR}^2 = \left(\frac{yz}{\epsilon}, \frac{xz}{\epsilon}, \frac{xy}{\epsilon}, -\frac{xyz}{\epsilon^2}\right) \begin{bmatrix} \sigma_x^2 & 0 & 0 & C_{x\epsilon} \\ 0 & \sigma_y^2 & 0 & 0 \\ 0 & 0 & \sigma_z^2 & 0 \\ C_{\epsilon x} & 0 & 0 & \sigma_\epsilon^2 \end{bmatrix} \begin{bmatrix} \frac{yz}{\epsilon} \\ \frac{xz}{\epsilon} \\ \frac{xy}{\epsilon} \\ -\frac{xyz}{\epsilon^2} \end{bmatrix}$$

$$\sigma_{DMAR}^2 = \left(\frac{yz}{\epsilon}\right)^2 \sigma_x^2 + \left(\frac{xz}{\epsilon}\right)^2 \sigma_y^2 + \left(\frac{xy}{\epsilon}\right)^2 \sigma_z^2 + \left(\frac{xyz}{\epsilon^2}\right)^2 \sigma_\epsilon^2 - 2\frac{xy^2z^2}{\epsilon^3}C_{x\epsilon}$$

It is worth testing the differences between the outcomes of the two error propagation equations and deciding whether the original one is retained or the latter one proposed here.

Line 131: It is worthwile adding that this relation is true if the bottom/top ages of a segment is given as negative ages in the past calculating from e.g 1950 BP, or b2k. If the ages are positive as usually used (e. g. $t_{bottom}$=23500 years, $t_{top}$=21000), then this does not hold. This can be easily fixed by changing the direction of the inequality signs.

Lines 236-246: The authors use a dry bulk density (DBD) value of 1.48 g/cm$^3$ for CLP loess, while 1.45 for others. For European loess, DBD is relatively well-defined (at least for East Central Europe, ECE): Újvári et al. (2010) [Quat.Sci.Rev. 29, 3157-3166] published a DRD value of 1.497+-0.079 g/cm$^3$ derived from 6 test sample measurements (2 methods), which was later confirmed by e.g. Peric et al. (2020) [Boreas 49, 841-857]. So, this value should be used for the ECE loess sites, and perhaps for others as well considering how close this is to the CLP loess DBD value. I suggest mentioning these studies here, just to provide a less CLP-biased overview of the topic.

Line 381, SBMAR equation: Why is this Eq (1)? We have seen several equations before in this manuscript. Those must be numbered and indicated as well, in my view.

14/09/2023
Gabor Ujvari

---

## Referee Comment (RC2)

[referee-annotated manuscript omitted]

---

## Author Comment (AC1)

**REFEREE #1: Gabor Ujvari**

In recent years, there has been a renaissance in the field of spatio-temporal reconstruction of sediment mass accumulation rates (MAR), probably due to the fact that atmospheric dust does indeed play a significant role in shaping Earth's climate, in both direct and indirect ways. Yet there are considerable uncertainties about the actual role of dust that need to be clarified in order to better understand how the Earth system works under present-day or past (e.g. glacial) climate conditions. Research on MAR reconstructions is taking several directions: 1) synthesis of new data from one or more type(s) of paleo-dust archives, 2) methodological innovations in the calculation of MAR, 3) database construction, and/or 4) focusing on periods that have not been included in reconstructions so far (e.g. Marine Isotope Stage 4-5-6). As such, a new database called ChronoLoess (Bosq et al.) is currently appearing in the ESSD pages, which can be extended in a standardised format. In contrast to ChronoLoess, which focuses on only one archive, the authors of this manuscript have compiled data from a large number of paleodust archives of different types (marine/terrestrial) in the Paleo±Dust database, following a common methodology. I consider the most important advance of the present paper to be that they have discussed the uncertainties associated with the dust mass accumulation rate (DMAR) calculations of each archive type and attempted to estimate them quantitatively, which in many cases is not a simple task. In addition, paleo-dust flux estimates from peat bog cores are included in the paleodust compilation for the first time.

Having read the manuscript, I find it to be neatly written, logical and easy to follow, and I consider it to be a significant step forward in the reconstruction and comparison of LGM and Holocene dust fluxes. The authors have tried to organise the database in a way that it can be used for future Earth system modelling, which is also a positive aspect. I have, of course, some critical comments on the manuscript, which I write about below (and in the annotated pdf file attached), but these do not affect the conclusions to any significant extent and are rather just a further refinement of the methodology. On the whole, the manuscript, with some modifications, can be published in ESSD.

Thank you very much for your comments. We have looked at each comment you have annotated in the pdf file, and incorporated changes in response to most of them in the submitted new text (with tracked changes). Next, we respond to one of these annotated comments, which we believe require a detailed explanation:

Annotated comment in section 2.7: "I don't want to increase the number of uncertainties that must be considered here, but there is a clear difference in the proportion of f10 of grain size distributions calculated using the Mie theory or the Fraunhofer approximation (see e.g. Figure 6 in Varga et al. 2019, Sedimentary Geology 389, 42-53). My guess is, however, that this uncertainty is  much smaller than the overall, conservative uncertainty (20%) on the <10 micron dust fraction and is therefore accounted for."

We have studied in detail the manuscript by Varga et al. (2019) that you mention, which has information that is very relevant to our manuscript. There, authors evaluate differences in the grain size distribution of a set of sediment samples using (a) different laser-diffraction devices, (b) different assumptions on the refractive index used in conjunction with Mie theory, and (c)

different theories (Mie theory vs. Fraunhofer approximation). Table 3 of Varga et al. (2019) nicely shows the effect of points (a) and (b), while Figure 6 shows the effect of point (c). We used the mean values in Table 3 for each device to calculate a 1-sigma uncertainty in the volume proportion of the <10-micron fraction, due to points (a) and (b) above. We do so by considering results for the clay, fine silt and medium silt results. We calculate the final 1-sigma uncertainty for each device by weighting the 1-sigma uncertainty of each one of these three size bins by the volumetric percentage. Given that medium silt is between 6.25-20 microns, we only consider 35.2% of the volumetric fraction of that size bin (see response to next comment). For example, for the HORIBA device, this calculation would result in:

[1.9vol% 61% + 12.8vol% 23% + 0.352 40.1vol% 14%] / (1.9vol% + 12.8vol% + 0.352 40.1 vol%)

= 21.1% (1-sigma uncertainty in $f_{10}$ for HORIBA due to uncertainty in optical settings).

This value is maximum for HORIBA (17.7% for FRITSCH, 8.4% for MALVERN), so we use this value to be conservative.

Figure 6 does not allow a similar treatment for the uncertainty associated with the choice of theory (Mie vs. Fraunhofer).

We combine this 21.1% uncertainty in terms of L2 norm (square root of sum of squares) with the 1-sigma uncertainty in $f_{10}$ that we had already discussed in the manuscript of 0.7% due to laser-diffraction reproducibility, which gives a final value of 21.1%, which we round to 20%. Also, we have decided to eliminate the 14% uncertainty associated with sieving prior to grain size measurements that we had previously discussed. This is because after re-inspection of the paper that evaluated this sieving error (Lopez-Garcia et al., 2021, doi:10.3389/fmars.2021.738479), we realized that the conclusions they reached are not applicable to our study. This is because they use sand-dominated sediment samples, and evaluate the relative error associated to manual sieving (compared to mechanical sieving) with 2000- and 125-µm meshes on the median grain size of the sieved sample between 125-2000 µm, and so this gives no information on how preliminary sieving at 62.5 or 125 µm (which is usually done prior to grain size measurements for dust-related studies) impacts $f_{10}$, which is what is actually of interest to us. Thus, the base uncertainty in f10 remains at 20% for laser diffraction measurements, although the source of this base uncertainty is different to what we previously described. This change is applied to the first paragraph of section 2.7:

"We assign the lowest uncertainty to $f_{10}$ when it is calculated from a full volumetric grain size distribution (Figure 3). This uncertainty is associated with the reproducibility of measurements (~0.7% after propagation of each bin's uncertainty), based on laser-diffraction determinations of 135 South American loess samples from three sites, dated between 8-53 ka BP (Coppo et al., 2022a), as well as with the use of different laser-diffraction devices and assumptions on their optical settings (19.9%), based on measurements of loess samples from East Central Europe (Varga et al., 2019). We thus assign a 20% relative uncertainty to $f_{10}$ when calculated from full grain size distributions using laser diffraction devices. Instead, Coulter counters measure particle volume more accurately than laser diffraction devices (Simonsen et al., 2018), so that volumetric grain size distributions and $f_{10}$, measured by this technique, are arguably more accurate as well.

We thus assign a lower relative uncertainty to $f_{10}$ of 5% when calculated from Coulter counter measurements. This specific value of 5% is, however, arbitrary as the authors are not aware of studies that quantified sources of uncertainty of Coulter counter-derived grain sizes. In many cases where such measurements are carried out, the full-size distribution data is not published and $f_{10}$ is not reported in the original study. For a number of Holocene sites published previously to 2015, Albani et al. (2015) compiled these distributions. Also, Albani et al. (2014) compiled $f_{10}$ values for Holocene and LGM dust archive sites published prior to 2014. For all these sites, $f_{10}$ was retrieved from these studies. For the rest of the sites in Paleo±Dust, if data was not available but the grain size distribution was plotted in the original study, we estimated $f_{10}$ visually adding an extra 10% of uncertainty (30% in total). To perform this visual estimation, we used a vector graphics editor (Adobe Illustrator) to draw two polygons: one that encompassed the area under the curve of the volumetric grain size distribution for all measured particle sizes, and one for sizes <10 µm. We calculated $f_{10}$ as the ratio of the latter to the former area. Greater uncertainty (30%) is associated to $f_{10}$ when it is calculated from reported grain size bin volumetric abundances (e.g., clay, silt), either because grain size was determined using the sieve and pipette method, or because the full grain size distribution is not reported. Also, a higher uncertainty is assigned when $f_{10}$ is calculated from a reported mean or median value (40%). To derive $f_{10}$ in these cases (from mean, median or bins) we used the average grain size distribution from the Coppo et al. (2022a) dataset to obtain linear least-square regression equations for $f_{10}$ vs. mean ($R^2 = 0.71$), $f_{10}$ vs. median ($R^2 = 0.81$), $f_{10}$ vs. $f_{20}$ ($R^2 = 0.88$), among others (see site-specific notes for more details), except when $f_{10}$ calculated from mean grain size was retrieved directly from Albani et al. (2014). When no grain size measurements are available for a given site, but only for near-by sites (<100 km) that are comparable in terms of their geomorphological setting, then the same value for $f_{10}$ is used in both sites, with an extra 10% uncertainty for the site with no data. The same is true for sites that include grain size data for a different time window than the one considered."

Line-by-line comments

Line 97, εDMAR: This is just a pedantic point, but the error of a quantity is usually denoted by the Greek sigma in statistics, I have never seen the use of epsilon for that purpose. What is the reason for using epsilon here? There is another reason of why I discourage the authors using epsilon: in dust provenance studies the Nd and Hf isotope compositions are often expressed in the epsilon notation (εNd, εHf), so this may potentially cause some misunderstandings.

We agree that "ε" is not the best choice, for the reason you mention. We are comfortable with using "σ", as in fact we report 1-standard deviation errors, and "σ" is usually used for that.

Lines 98-100: The authors presented uncertainty estimations for MAR and MAR10, which disregard potential correlations between the variables in the MAR equations, so these are not full statistical treatments. I suggest the error propagations below.

Let us rewrite the DMAR equation: $DMAR = \frac{h_{thick} \times DBD \times EC}{t_{bottom} - t_{top}} = \frac{h_{thick} \times DBD \times EC}{\Delta t} = \frac{x \times y \times z}{\epsilon}$ ($\epsilon = \Delta t = t_{bottom} - t_{top} = t_b - t_t$)

The errors (or variances, or standard deviations) of x, y, z and $\epsilon$ ($\sigma_x^2 = s_{h_{thick}}^2$, $\sigma_y^2 = s_{DBD}^2$, $\sigma_x^2 = s_{EC}^2$, $\sigma_\epsilon^2 = s_{\Delta t}^2$) are known. Obviously, x ($h_{thick}$)and $\epsilon$ ($\Delta t = t_{bottom} - t_{top}$) are correlated ($C_{x\epsilon} \neq 0$, $C_{x\epsilon}$ is the covariance of x and $\epsilon$, and $C_{x\epsilon} = \varrho_{x\epsilon}\sigma_x\sigma_\epsilon$, where $\varrho_{x\epsilon}$ is correlation of x and $\epsilon$), while no correlation exists between the other variables: $C_{xy} = C_{yz} = C_{y\epsilon} = C_{z\epsilon} = 0$. DBD and EC may be correlated, this can be a matter of debate, but we suppose zero correlation here. With this, the uncertainty of DMAR is given by:

$$\sigma_{DMAR}^2 = f^* \times M \times f$$

, with $f^* = \left(\frac{\partial DMAR}{\partial x}, \frac{\partial DMAR}{\partial y}, \frac{\partial DMAR}{\partial z}, \frac{\partial DMAR}{\partial \epsilon}\right)$, $f = \left(\frac{yz}{\epsilon}, \frac{xz}{\epsilon}, \frac{xy}{\epsilon}, -\frac{xyz}{\epsilon^2}\right)$, $M = \begin{bmatrix} \sigma_x^2 & C_{xy} & C_{xz} & C_{x\epsilon} \\ C_{yx} & \sigma_y^2 & C_{yz} & C_{y\epsilon} \\ C_{zx} & C_{zy} & \sigma_z^2 & C_{z\epsilon} \\ C_{\epsilon x} & C_{\epsilon y} & C_{\epsilon z} & \sigma_\epsilon^2 \end{bmatrix}$

$$\sigma_{DMAR}^2 = \left(\frac{yz}{\epsilon}, \frac{xz}{\epsilon}, \frac{xy}{\epsilon}, -\frac{xyz}{\epsilon^2}\right) \begin{bmatrix} \sigma_x^2 & 0 & 0 & C_{x\epsilon} \\ 0 & \sigma_y^2 & 0 & 0 \\ 0 & 0 & \sigma_z^2 & 0 \\ C_{\epsilon x} & 0 & 0 & \sigma_\epsilon^2 \end{bmatrix} \begin{bmatrix} \frac{yz}{\epsilon} \\ \frac{xz}{\epsilon} \\ \frac{xy}{\epsilon} \\ -\frac{xyz}{\epsilon^2} \end{bmatrix}$$

$$\sigma_{DMAR}^2 = \left(\frac{yz}{\epsilon}\right)^2 \sigma_x^2 + \left(\frac{xz}{\epsilon}\right)^2 \sigma_y^2 + \left(\frac{xy}{\epsilon}\right)^2 \sigma_z^2 + \left(\frac{xyz}{\epsilon^2}\right)^2 \sigma_\epsilon^2 - 2\frac{xy^2z^2}{\epsilon^3}C_{x\epsilon}$$

It is worth testing the differences between the outcomes of the two error propagation

equations and deciding whether the original one is retained or the latter one proposed here.

We believe this error propagation would be correct in the case that we considered various sequential vertical sections (each with its own $h_{thick}$ and $\Delta t$) at each site. However, in our case we calculate only one DMAR value at each site (one for Holocene and one for LGM) and therefore for each site we only have one $h_{thick}$, one $\Delta t$, one DBD and one EC value to calculate our DMAR (one set of these values to calculate one Holocene DMAR, and another set of these values to calculate one LGM DMAR). Therefore, we believe that the covariances between the various variables within the equation are automatically zero.

Line 131: It is worthwhile adding that this relation is true if the bottom/top ages of a segment is given as negative ages in the past calculating from e.g 1950 BP, or b2k. If the ages are positive as usually used (e. g. $t_{bottom}$=23500 years, $t_{top}$=21000), then this does not hold. This can be easily fixed by changing the direction of the inequality signs.

In fact, we believe the direction of the inequality signs is correct. Let's use two examples to show this. The first is an example where the DMAR is not discarded: $t_{bottom}$=24000 ± 500 years, $t_{top}$=21000 ± 1000 years (each age does not fall within the other age's uncertainty: good scenario). In this case, if we use the inequalities as written in the manuscript, $t_{bottom} = 24000 \leq t_{top} + \sigma t_{top} = 21000 + 1000 = 22000$, which is not true, and $t_{top} = 21000 \geq t_{bottom} - \sigma t_{bottom} = 24000 - 500 = 23500$, which is also not true. That is, neither of the inequalities hold, and thus, the DMAR value is not discarded (which is what we want). Now, the second example is $t_{bottom}$=24000 ± 3500 years, $t_{top}$=21000 ± 4000 years. This is an example where both ages are within the

uncertainty of the other age. This is the bad scenario, for which we discard the DMAR value. If we replace as before, we see that both inequalties hold. Thus, we discard the DMAR value.

Lines 236-246: The authors use a dry bulk density (DBD) value of 1.48 g/cm$^3$ for CLP loess, while 1.45 for others. For European loess, DBD is relatively well-defined (at least for East Central Europe, ECE): Újvári et al. (2010) [Quat.Sci.Rev. 29, 3157-3166] published a DRD value of 1.497+-0.079 g/cm$^3$ derived from 6 test sample measurements (2 methods), which was later confirmed by e.g. Peric et al. (2020) [Boreas 49, 841-857]. So, this value should be used for the ECE loess sites, and perhaps for others as well considering how close this is to the CLP loess DBD value. I suggest mentioning these studies here, just to provide a less CLP-biased overview of the topic.

Thanks for pointing this out. We agree with further refining the assumptions done on loess DBD when it is not measured. Please, check the changes we have introduced to the first sentence of the second paragraph of section 2.5:

"If *DBD* was not measured and no near-by sites with measured density are available, a value for *DBD* of 1.45 g cm$^{-3}$ and an uncertainty of 30% are used for loess and lake sites, except for loess sites in the Chinese Loess Plateau and in East Central Europe (west of 21.5ºE) where mean values of 1.48 g cm$^{-3}$ (Kohfeld and Harrison, 2003) and of 1.497 g cm$^{-3}$ (Újvári et al., 2010; Peric et al., 2020) are preferred, respectively."

Line 381, SBMAR equation: Why is this Eq (1)? We have seen several equations before in this manuscript. Those must be numbered and indicated as well, in my view.

The reason why we start the numbering with this equation is that this is the first equation that is explicitly mentioned in the text. All the previous equations are not, so we believe it is not necessary to number them, although this will ultimately be decided by the typesetting of the journal.

**ANONYMOUS REFEREE #2**

I have thoroughly reviewed the manuscript titled "Paleo±Dust: Quantifying uncertainty in paleo-dust deposition across archive types" by Nicolás J. Cosentino and co-authors. The primary objective of this manuscript is to compile updated bulk and <10-µm paleo-dust deposition rates across various archive types between 2016 and 2022, referred to as the Paleo±Dust dataset. This dataset encompasses a total of 284 pre-industrial Holocene (pi-HOL) and 208 Last Glacial Maximum (LGM) dust flux records. Based on the Paleo±Dust dataset, the manuscript calculates that the global LGM:pi-HOL ratio of <10-µm dust deposition rate is $3.1 \pm 0.8$ (1σ), which is a critical parameter for future paleodust studies and simulations using Earth system models of high to intermediate complexity.

In my opinion, this dataset presented in the manuscript are novel and hold great potential being useful in the future. The materials and methods are described in detail y, and the dataset is

complete and accessible through the provided identifier. The dataset is currently usable in its existing format and size. Additionally, the manuscript is well-structured, nicely illustrated, and well-written. However, I have a few suggestions that require further clarification:

Thanks for your comments.

Uncertainty in the Mass Fraction of Dust: It is acknowledged that most organic matter in loess is post-depositional in origin, it's important to note that carbonate in loess primarily originates from dust source areas and constitutes a significant portion (approximately 14wt%) of dust materials (Meng et al., 2019, Geophysical Research Letters 46, 4854-4862). After deposition, carbonate minerals in loess can undergo redistribution in the soil through processes such as dissolution, migration, and reprecipitation (Meng et al., 2015, Geophysical Research Letters 42, 10,391-310,398, and Meng et al., 2018, Earth and Planetary Science Letters 486, 61-69). Previous research on the Chinese Loess Plateau (Meng et al., 2015, Geophysical Research Letters 42, 10,391-310,398, and Meng et al., 2018, Earth and Planetary Science Letters 486, 61-69) and Central Asia (Zhang et al., 2023, Catena 232, 107420) has indicated that wet climates in interglacial periods (e.g., Holocene) tend to deplete carbonate in the soil, whereas carbonate loss during the last glacial period is limited. Therefore, it would be more reasonable to assume that 10±5wt% of carbonate was lost in the Holocene and none in the last glacial period. Similarly, for organic matter, it's essential to consider its varying content between the Holocene (high) and the last glacial period (low), as supported by some studies (e.g., Yang et al., 2015, PNAS, 112, 13178-13183). Based on previous works, it would be more reasonable to assume that TOC content was 1±0.5wt% for the Holocene and 0.2±0.2wt% for the last glacial period.

We acknowledge that our treatment of carbonates in loess in regards to its origin (and how it should be considered when calculating EC) was lacking. We have thoroughly modified the way we calculate EC in loess-paleosol sequences to reflect many of the ideas you mention here, even when for most sites changes in DMAR due to these modifications are small. We explain these modifications in section 2.6.1 of the manuscript:

"In loess studies with a focus on dust dynamics, *EC* is usually assumed to be 1, that is, loess is assumed to be fully composed of aeolian dust. However, while organic particles present in dust sources may be transported by wind and later deposited in the same manner as lithic particles (Muhs et al., 2014), here we assume that organic matter in loess is post-depositional in origin (e.g., Hatté et al., 2001), and subtract total organic carbon (TOC) to calculate *EC*. Instead, carbonates constitute in some regions a significant fraction of airborne dust (Scheuvens and Kandler, 2014), which supports the interpretation that carbonates in loess are dominantly primary, that is, derived from dust sources (e.g., Li et al., 2013; Meng et al., 2019). However, carbonates in loess have also been shown to be authigenic (e.g., Da et al., 2023). Unfortunately, for most published studies it is not possible to calculate the contributions of primary and authigenic carbonates to the total carbonate content (TCC) of loess. We thus assume that 50% of carbonate present in loess in primary, while 50% is authigenic. Furthermore, previous research has shown that wet climates during interglacials tend to deplete 10% of carbonates in soils (by weight), while no loss occurs during glacial periods (Meng et al., 2015, 2018; Zhang et

al., 2023). We account for this post-depositional loss of carbonates in Holocene loess-paleosol sections in our *EC* calculations.

Relative 1-σ uncertainty in *EC* is assumed to be 10% (20%) if both (either) TOC and (or) TCC are (is) reported (Figure 2). If both TOC and TCC are available and there is also a quantification of volcanic inputs, then the relative uncertainty is reduced to 1%. Instead, when neither TOC nor TCC data is available, loess-paleosol units are classified as either organic carbon-rich or organic carbon-poor, based on the physical description of the unit of interest, and the sum of TOC and TCC is assigned a value of 6 wt% and 2 wt%, respectively. These correspond respectively to the first and third quartiles of the sum of TOC and TCC for sites in Paleo±Dust where both TOC and TCC were determined (N = 28). In addition, when TOC is not determined, values of 0.2 wt% and 1 wt% are assumed for LGM and pi-HOL sites, based on previous studies (Yang et al., 2015). In these cases when both TOC and TCC are assumed, relative uncertainty in *EC* is highest at 30%."

Minor Comments:

This manuscript is updated on the basis of Albani et al., (2015), if the updated sites are marked in Figure 5, it is easier to make the reader clear about the value of this dataset.

In fact, our manuscript is updated not only on the basis of Albani et al. (2015), but also on the basis of other compilations that came both before (Albani et al., 2014) and after (e.g., Kienast et al., 2016) that one. It is true that one important aspect of our compilation is the new sites that are first compiled in it, and we discriminate these sites in the tabulated files. We believe there is no simple way to include the information of "new site" vs. "previously published site" in this figure, given the information that is already in it, and still make it readable. We would thus prefer not to modify this figure in this sense.

It should be noted that the last glacial-interglacial period corresponds to marine isotope stages (MIS) 2-5, and not to the Holocene (MIS 1) or the last glacial period (MIS 2-4). The present interglacial, which is the Holocene, is ongoing interglacial period, not last interglacial.

We agree. We have changed the text to reflect this in sections 2.2.1 and 2.3:

(section 2.2.1) "One of the main objectives of Paleo±Dust is to compile global constraints on *DMAR* for the ongoing interglacial and last glacial periods."

(section 2.3) "They found a high absolute error during the ongoing interglacial and last glacial periods of approximately ±6 kyr, which remained relatively constant during this time span."

Lines 175-176 : If dating uncertainty is not reported, 3.4%, 9.1% and 13.1% relative uncertainties are assumed for 14C-, OSL- and thermoluminescence-based age determinations. Please give the references for the uncertainties. In my understanding, the relative uncertainties associated with these dating methods are hardly accurate to one decimal point as the values mentioned (e.g., 9.1%). For instance, it is commonly accepted that the relative uncertainties in OSL dating method are often estimated to be around 10%.

What we did to derive these relative uncertainty values is explained in the same paragraph where these values are mentioned (first paragraph of section 2.3): "If dating uncertainty is not reported, 3.4%, 9.1% and 13.1% relative uncertainties are assigned to $^{14}$C-, OSL- and thermoluminescence-based age determinations. These values represent the 75th percentile of the distributions of measured $^{14}$C (N = 83), OSL (N = 129), and thermoluminescence (N = 20) age relative uncertainties for sites included in Paleo±Dust."

Consider adding the dust data of Sihailongwan maar Lake in northeastern China (Zaarur, S., Stein, M., Adam, O., Mingram, J., Liu, J., Wu, J., Raveh-Rubin, S., Erel, Y., 2020. Synoptic stability and anomalies in NE China inferred from dust provenance of Sihailongwan maar sediments during the past ~80 kyr. Quaternary Science Reviews 239, 106279). This could contribute valuable data to the manuscript.

After carefully analyzing the information reported in the paper you mention, as well as in Mingram et al. (2018, doi:10.1016/j.quascirev.2018.09.023), we decided to include this site on our dataset. Thanks for pointing it out.

Overall, the dataset shows promise and provides valuable contributions to the field of paleo-dust research, but addressing the above suggestions will further enhance its quality and impact.

Thanks for your comments.